# COVID Variants, Villain and Victory: A Bioinformatics Perspective

**DOI:** 10.3390/microorganisms11082039

**Published:** 2023-08-09

**Authors:** Nityendra Shukla, Neha Srivastava, Rohit Gupta, Prachi Srivastava, Jitendra Narayan

**Affiliations:** 1CSIR Institute of Genomics and Integrative Biology, Mall Road, Delhi 110007, India; nitinshukla218@gmail.com (N.S.); rohit.gupta.delhi1995@gmail.com (R.G.); 2Amity Institute of Biotechnology, Amity University, Uttar Pradesh, Lucknow Campus, Lucknow 226010, India; ns011982@gmail.com (N.S.); psrivastava@amity.edu (P.S.)

**Keywords:** SARS-CoV-2, genomics, immunoinformatics, drug design, bioinformatics, COVID-19

## Abstract

The SARS-CoV-2 virus, a novel member of the Coronaviridae family, is responsible for the viral infection known as Coronavirus Disease 2019 (COVID-19). In response to the urgent and critical need for rapid detection, diagnosis, analysis, interpretation, and treatment of COVID-19, a wide variety of bioinformatics tools have been developed. Given the virulence of SARS-CoV-2, it is crucial to explore the pathophysiology of the virus. We intend to examine how bioinformatics, in conjunction with next-generation sequencing techniques, can be leveraged to improve current diagnostic tools and streamline vaccine development for emerging SARS-CoV-2 variants. We also emphasize how bioinformatics, in general, can contribute to critical areas of biomedicine, including clinical diagnostics, SARS-CoV-2 genomic surveillance and its evolution, identification of potential drug targets, and development of therapeutic strategies. Currently, state-of-the-art bioinformatics tools have helped overcome technical obstacles with respect to genomic surveillance and have assisted in rapid detection, diagnosis, and delivering precise treatment to individuals on time.

## 1. Introduction

The COVID-19 pandemic, an unprecedented catastrophe, affected nearly every country on multiple levels. Since its emergence in December 2019, the pandemic has caused severe loss of human life, damage to healthcare and educational systems, and displaced economies worldwide. The scientific research community currently faces an arduous challenge since the virus undergoes frequent mutations, leading to the emergence of new variants. The high infectivity and lethality rates of the SARS-CoV-2 virus during its initial days and now, with the given propensity of the virus to mutate, further cause it to remain a major concern. To monitor and track changes in the SARS-CoV-2 genome caused by mutations and identify novel variants, various sequence-based surveillance, laboratory, and epidemiological investigations are now regularly being conducted. Continuous and vigilant genomic surveillance and analysis of viral variants in real time are crucial in the development of effective diagnostics, therapeutics, and vaccines to combat the onslaught of the pandemic.

A variant is defined as having two or more mutations that distinguish it from other variants in terms of spreading. In collaboration with the Centers for Disease Control and Prevention (CDC) and the US Department of Health and Human Services (HHS), the two institutions established an Interagency Group (SIG) to develop a classification system for SARS-CoV-2 variants. They classified them into four categories and marked them as follows: (1) variants of interest (VOI), (2) variants of concern (VOC), (3) variants of high consequences (VOHC), and 4) variants being monitored (VBM) [1,2]. Variants of interest (VOI) are variants with specific heredity marker changes characterized by alterations in the receptor-binding domain (RBD). In addition to reduced antibody neutralization generated post-infection and inoculation, variants can result in decreased efficacy of therapeutics, possible analytic effect, or anticipated expansion in contagiousness or chronic illness. During different peaks of the pandemic, certain variants emerged as notable players. In the spring of 2021, the Iota variant (B.1.526) took the spotlight in the United States, while in the summer of the same year, the Epsilon variant (B.1.427 & B.1.429) gained prominence in the region, exhibiting an approximate 20% increase in transmission rate. In the United Kingdom (UK), the Eta variant (B.1.525) emerged during the spring of 2021. In India, the Kappa variant (B.1.617.1 & B.1.617.3) made its presence known, eventually giving rise to the highly infectious Delta variant. As of June 2023, the VOIs circulating in the population are XBB.1.5 and XBB.1.16, both of which are suspected to possess increased immune escape potential and continue to cause surges in infection rates (Figure 1) [3].

Lineages classified as VOC have been the most devastating during the course of the pandemic, leading to high hospitalization and mortality rates, in addition to the development of chronic health issues such as post-acute sequelae of COVID-19 (PASC), more commonly known as long COVID, type 2 diabetes, fatigue, and dyslipidemia, among others, resulting in lifelong disability and medical care [5]. The hallmarks of VOCs are their increased rates of transmission and immune escape potential, leading to a significant reduction in neutralization antibodies in addition to decreased effectiveness of therapeutics and vaccines, resulting in high rates of hospitalization and death. The major VOCs throughout the course of the pandemic included the Alpha (B.1.1.7) variant, identified in the UK in December 2020 [6], possessing 23 mutations as compared to the wild-type variant, with an extremely high transmissibility rate of ~70% [7]; it had become dominant in 21 countries by March 2021. The Beta (B.1.351) variant was first identified in South Africa in October 2020, defined by 12 mutations in its genome compared to the wild-type, and was the first VOC to display decreased vaccine efficacy and monoclonal antibody resistance [8]; it was responsible for pandemic waves across the African continent. The Delta (B.1.617.2), first identified in India in October 2020, was the most devastating out of all VOCs so far. It was considered to be 50% more transmissible than the Alpha variant and was the dominant lineage during India’s second wave, and it contributed to third waves in the UK and South Africa and eventually became the dominant variant worldwide. The Delta variant possessed 13 mutations, 8 of which were in the spike (S) protein [9], thus increasing its transmissibility and ability to infect the lower respiratory tract, causing severe disease conditions such as acute respiratory distress syndrome (ARDS) [10]. The current VOC, Omicron (B.1.1.529), identified in November 2021 in South Africa, is the most diverse variant of all the VOCs before it, carrying approximately 50 mutations, and has since continuously evolved into subvariants, leading to WHO changing their classification system; thus, Omicron subvariants are tracked independently [11]. Omicron and its subvariants possess significant immune escape potential, with Omicron subvariants such as BA.1, BA.2, BA.4, and BA.5 being poorly neutralized by first-generation vaccines, in addition to being resistant against most monoclonal antibodies except bebtelovimab [3]. However, Omicron seems to display a bias for infecting the upper respiratory tract over the lower respiratory tract [12], possibly due to lower fusogenicity and possibly altered tissue tropism due to its ability to be primed by endosomal proteases, such as cathepsins, a phenomenon seen in SARS-CoV [13,14,15]. This mechanism is hypothesized to somewhat play a role in the decreased severity of disease, though all current evidence that supports this is limited to rodent models [15,16].

Thus, these variants play the role of a villain, causing serious health complications in people who are infected and post-infection as well. Delta variants have been found to be significantly more virulent as well as contagious as compared to other variants and were mainly responsible for the global surges in 2021, particularly in India, South Africa, and the United Kingdom. The high transmission rates expose the population to repeat infections, potentially leading to a higher risk of developing long COVID [17], whose symptoms can last for years and in some cases, lifelong. Additionally, repeat COVID-19 infections heighten the risk of developing adverse comorbidities such as cardiovascular disease, type 2 diabetes [18], or myalgic encephalomyelitis/chronic fatigue syndrome (ME/CFS) [19]. Furthermore, the Delta variant displayed increased lethality as compared to the Omicron variant, with a median case-fatality rate (CFR) of 8.56 as compared to 3.04, respectively [20], probably due to increased vaccine coverage and decreasing pathogenicity as the virus continues to evolve. As the virus evolves, the effect of medical therapeutics and available interventions continues to decrease, leading to an increased risk of hospitalization, particularly in high-risk groups. New mutations drive a strong increase in transmission rates, and a constantly evolving machinery of SARS-CoV-2 means decreased drug response to new symptoms that the virus may give rise to. Therefore, mutations and variants impeded research; as a result, universal inoculation was the effective long-term solution to thwart the growing risk of an evolving SARS-CoV-2 virus and assist in decreasing mortality and hospitalization rates [21].

Bioinformatics has created a milestone and has played an essential role in COVID-19 research since the emergence of the pandemic [22], especially in detecting and tracking new variants, as well as in allowing for open data-sharing across the world. Wide varieties of bioinformatics tools and techniques have been used successfully in the interpretation of the genomic architecture of SARS-CoV-2 and its variants, as well as in the creation of mathematical models used to predict infection spread, design containment methods, and drive public health policy decisions [23,24]. They have been utilized in analyzing the data generated from genomics, transcriptomics, proteomics, and structural omics, as well as single-cell data. They have contributed to a better understanding of the underlying molecular mechanisms of viral pathogenesis, allowing for the rapid identification of vaccine and drug candidates [25], particularly for the rapid deployment of repurposed drugs during the pandemic [24,26] for clinical research and eventually, for clinical trials. From this perspective, we highlight the strong contributions and achievements of bioinformatics tools and techniques in combating the COVID-19 pandemic.

## 2. Tracking of SARS-CoV-2

Next-generation sequencing (NGS), also defined as high-throughput sequencing technology, has become a widely adopted approach in genomics research to identify virus origins, delineate mechanistically phenotypic causative pathophysiology, and elucidate genotypic ramifications in infected individuals. These technologies have further led to the development of novel bioinformatics tools, pipelines, algorithms, and machine-learning approaches that have now become robust practices in understanding the virus genome [27].

Advancements made in metagenomics have aided in the identification of plausible co-occurring pathogens that may play a role in the clinical outcome of SARS-CoV-2 infections. Thus, studying host–pathogen interactions also become important in this context. Augmenting this with different analytical strategies that bioinformatics tools offer allows us to unravel the same genomic data via different lenses to uncover the underlying layers of host–pathogen interactions, which may modulate disease severity as well as clinical outcome [28]. Various metagenomics tools such as Kraken2 [29], MOTHUR [30], and QIIME2 [31] provide deep insights into genomic data as well as provide ways to visualize data; thus, meaningful insights can be extracted. Further, this is combined with Seurat [32], a single-cell analysis R library, which allows for spatial visualization of multiple clusters of cells that may themselves be strongly implicated in disease pathophysiology. The analysis is then extended further with downstream machine-learning tools, such as singleR [33] and clustifyr [34]. These can be leveraged to annotate cell clusters with public data libraries with high confidence and further identify causal gene–pathway relationships with respect to the cell types that may be contributing factors in disease pathophysiology.

In this context, an opportunity exists to develop a cohesive and well-developed workflow that takes into account distinct steps that occur in the data analysis, thus streamlining the process of variant discovery. Typically, an NGS data analysis pipeline consists of various integral steps, such as quality control of the data, deletion of extraneous host data, and read assembly, followed by taxonomic classification (of pathogens) and, lastly, supplemented by virus genome verification. Due to the wide adoption, use, and build of various open-source methods, various tools exist that help in further carrying out each step elucidated above.

Bioinformatics tools have accelerated efforts in unwinding the mutations and genetic variation of the SARS-CoV-2 virus. Undoubtedly, SARS-CoV-2 has the potential to adapt during the current pandemic because of its high and rapid mutation rate. How this evolution has an impact on the transmission, duration, and gravity of disease is still under study. The increasing amount of evidence of human–wildlife interactions has facilitated the zoonotic transmission from animals to humans.

In SARS-CoV-2, the spike protein of the virus has sufficient binding affinity with the receptor of angiotensin-converting enzyme 2 (ACE2), which is crucial for host cell entry and human infection. Several computational models and data have also indicated that additional mutations at the binding site strengthen the binding affinity [35,36]. According to Nextstrain [37], SARS-CoV-2 underwent roughly one genetic change per week based on the substitution rate. The mutation rate of the SARS-CoV-2 virus is estimated at 1 × 10^−6^–2 × 10^−6^ mutations per nucleotide per replication cycle [38]. The availability of abundant data and resources on the viral genome exhibited the intense response to the pandemic by tracking and tracing of infection to drug and vaccine development. Nextstrain collects, analyzes, visualizes, and maintains metadata of SARS-CoV-2 from various public data repositories, including NCBI (www.ncbi.nlm.nih.gov, accessed 9 July 2023), GISAID (www.gisaid.org, accessed 9 July 2023), and GitHub repositories.

The genomic data allows us to understand the dynamics of the genome evolution of viruses, including mutations and natural selection, which are very important aspects for detecting clinical variants and their epidemiological significance, understanding viral evolution and responses of the immune system to mutations, and vaccine and drug development. The graphical display of sequence data is also available for a better understanding of the genomic epidemiology of SARS-CoV-2 (Figure 2). Bioinformatics strategies play a key role in the rapid detection, tracing, understanding, analysis, evaluation, and treatment of COVID-19. With the sheer amount of available data, scientists are globally investigating evolution at the genome and protein level to tackle the pandemic [39]. For early detection, tracking, tracing, sequencing, and the creation of therapeutic methods, a variety of bioinformatics workflows and tools have been developed. Interestingly, we have seen a more than four-fold increase in drug designing tools and software in the last few years, while ML tools have increased by two-fold (Figure 3). To avoid the false-positive and false-negative detection of qRT-PCR [40], a computation-based primer PriSeT has been developed for detecting the specificity and sensitivity of the qRT-PCR test [41].

Next-generation sequencing workflow V-Pipe [44] and Haploflow [45] are developed to evaluate and monitor genetic diversity and mutation. Amplicon-based metagenomics sequencing has been prevalent throughout the pandemic and has been extensively used to study sequence divergence [46,47]. Several open-source databases, such as GISAID, NCBI, and EMBL-EBI, provide easy access and submission for high-quality SARS-CoV-2 genome sequence metadata globally. For the detection and annotation of viral genomes, numerous bioinformatics tools, software, and pipelines have been developed, including VADR [48], VBRC tools (https://4virology.net/virology-ca-tools/, accessed 4 July 2023), and VIRULIGN [49]. Various databases, such as UniProt, Pfam, and Rfam [50], are available to allow users to investigate coding and non-coding sequence evolution and diversity to understand viral epidemiology and evolution.

Besides this, numerous platforms and models were developed to study the fundamentals of evolutionary, epidemiology, and phylogenetic divergence of viruses. The phylodynamic models include BEAST 2 [51] for phylogeographic reconstruction; epidemic-mathematical models such as COPASI [52] and COVIDSIM [53]; and evolutionary tracking models like CoV-GLUE [54] and CoVe-Tracker are available [55]. The machine-learning model Covidex [56] is an open-source alignment tool based on Nextstrain and GISAID data used for the rapid classification of viral genomes in pre-defined clusters isolated from the population. Pangolin [57] rapidly assigns the most likely classification for a large number of genome sequences, which is particularly useful for local and global surveillance. All of these tools are freely available around the world for users. Thus, bioinformaticians are quickly responding to the pandemic and have provided easy and rapid access to COVID-19-specific tools and techniques for tracking and tracing the virus.

Certain limitations do currently exist in the space of NGS; bottlenecks continue to plague the efficacy of bioinformatics tools that could otherwise be leveraged to their full potential. More often than not, the virus reads (e.g., SARS-CoV-2 reads) are usually low in genomic data; this extends to single-cell data as well. In this case, to perform detailed analysis, different approaches need to be utilized; for instance, a homology-based approach or a protein structure-based method. Furthermore, robust and time-sensitive analysis becomes a challenge with the ever-increasing magnitude of availability of full-length genome sequences of SARS-CoV-2, spanning hundreds of gigabytes. One approach is to randomly subsample data and analyses rather than performing a full phylogenetic analysis on the entire genome data, which would be time-consuming and produce extremely complex results that may not lead to mechanistic insights critical to understanding virus transmission mechanisms. Thus, it is critical to continue developing bioinformatics tools with long-term strategic applications in mind so that researchers can perform efficient and rapid data analysis and clinical trial deployment.

## 3. Vaccine Status and Development

At least eighty infectious agents that manifest on a regular basis are known to be pathogenic in humans. Out of these eighty, more than thirty have licensed individual vaccines that target 26 of these infectious diseases—most of which are either viral or bacterial in nature. Given that vaccines have been instrumental in reducing mortality rates across the world at tremendous rates, it is also important to note that some of these vaccines are being used regularly and have been deployed to primarily circumvent childhood infections. Vaccination programs have been instrumental in reducing morbidity and mortality and inferring immunity to immunocompromised individuals and to children [58].

SARS-CoV-2 rapid evolution and transmission that continued unabated due to lack of herd immunity and lack of clinical drugs that could provide effective treatment meant that a mass vaccination program needed to be dispositioned immediately. Bioinformatics databases—which contain data with respect to nucleic acid sequences, protein sequences, ontologies, host databases, pathogen databases, and functional immunological databases (to name a few)—played a pivotal role in accelerating the development of novel mRNA vaccines that could shift the momentum that SARS-CoV-2 infection was gaining at a rapid pace. Further, the combined use of bioinformatics tools and machine-learning tools to make use of these databases was further monumental in guiding the precise development of the vaccines [38]. Further, accelerated research in these areas proved instrumental in the development of candidate vaccines (Figure 3). A variety of AI modalities were utilized for effective vaccine development and for evaluating the safety of such vaccines. A few of the vaccines that gained emergency use approval during the pandemic included: BNT162b2 from Pfizer-BioNTech, mRNA-1273 from Moderna, ChAdOx1 nCoV-19 from Oxford-AstraZeneca, and JNJ-78436735 from Johnson and Johnson [59].

Multipronged in silico approaches, including bioinformatics, vaccino-genomics, immunoinformatics, structural biology, and molecular dynamic simulations, are freely available to support precise vaccine design (Table 1). The sequenced SARS-CoV-2 genome is widely used in computational tools and databases for predicting novel B-cell and T-cell epitopes in vaccine development, immunity protein analysis, and immunization modeling. It deserves to be mentioned that immunoinformatics continues to be challenged by the accurate prediction of B-cell epitopes (BCE) and T-cell epitopes (TCE), which form the basis of the development of epitope-based vaccines [60].

During early March 2020, when the pandemic was gaining a foothold, a structure-based immunoinformatics methodology was employed to determine epitopes in SARS-CoV-2 for potential peptide-based vaccine design initiation. Wang and colleagues predicted 9 highly antigenic B-cell epitopes on the Spike (S) protein as well as 62 T-cell epitopes [61]. Reverse vaccinology (RV) is also a broadly used technique that employs computational methods to identify open reading frames (ORFs) from viral genomes, resulting in the discovery of new antigens [62]. The VaxiJen [63] server is used to study the physicochemical properties of antigen epitopes. Various machine-learning, as well as deep-learning models, have been employed to predict novel immunogenic subunits. These models have been crucial in identifying the epitopes for cytotoxic and helper T lymphocytes as well as in exploring the genetic polymorphism in the target human population [64,65]. Structural vaccinology techniques play a prominent role in identifying structurally stable, safe, and potent peptides as novel vaccine candidates [66]. Current bioinformatics tools such as Sprint [67], ModIAMP [68], pepATTRACT [69], PEPFOLD3 [70], IEDB [71], MHCPRED [72], SVMtrip [73], and AllerTop [42] have now been integrated into the programs for vaccine development.

It has to be noted, though, the drop in values across all categories in 2022 is due to a multitude of factors, such as a significant decrease in publications due to delays in peer-review processes and timelines due to a lack of manpower/staff, since an overwhelming number of COVID-19-related studies were submitted. Furthermore, financial resources were diverted from genomic sequencing to more clinical measures of control for the pandemic—such as support for healthcare infrastructure, vaccine development, and medical supplies/equipment costs to mitigate impact [74,75]. The future of immunoinformatics is important, as it will determine the pace with which we will overcome the next pandemic, in case it manifests. The rapid design, development, and deployment pipelines for vaccines will need to be refined with iterative use, reuse, and refinement of existing open-source bioinformatics databases and tools, as well as the development of new tools/pipelines that will be created in the future. Thus, bioinformatics shows a promising role in the rapid development of potent vaccine candidates against SARS-CoV-2, fighting the global mortality of COVID-19. 

## 4. Discussion and Conclusions

Finally, a major milestone in the fight against the pandemic was reached on 5 May 2023, when the Director-General of the World Health Organization (WHO) declared that COVID-19 was no longer classified as a ‘global health emergency’ but rather an ongoing health concern [76]. While this announcement marked a significant achievement, it is important to acknowledge that SARS-CoV-2, the virus responsible for COVID-19, continues to undergo rapid evolution, adapting to its environment and potentially enhancing its ability to evade immune responses and spread. However, there are indications that these changes in the virus may come at the expense of its severity. The development of effective vaccines in record time has played a major role in modifying the evolutionary landscape and weakening the severe effects of the virus. While predicting the evolutionary path of the virus is still hard, consistent genomic surveillance will allow for early prediction of its transition. Vaccines and therapeutics still need to be updated to maintain immunity in the population and prevent hospitalization and death, especially in high-risk groups and children, while also preparing to put systems in place so that states are better prepared for future pandemics and global health emergencies.

Bioinformatics has played a transformational role in SARS-CoV-2 diagnosis and treatment through high-quality, well-curated pipelines, software, and datasets, allowing for rapid design, development, and deployment. These are important not only in drug discovery but also in drug development and allied processes. It entails using informatics to gain new insights into health and disease, managing data during clinical trials, and reusing clinical data. We discovered that bioinformatics tools were critical in bolstering efforts to combat the COVID-19 pandemic, identifying and informing the public about emerging variants and their potential phenotypic manifestations and analyzing the virus at both the sequence and structural levels, and resulting in the rapid development of vaccines and other effective therapeutics to combat severe complications and, more importantly, develop vaccines. Thus, bioinformatics will continue to play a pivotal role in shaping the next paradigm of health, advancing personalized medicine and pushing the boundaries of elucidating the genomic architecture of infectious diseases, and ensuring we emerge victorious in safeguarding public health against unprecedented infectious diseases.

## Figures and Tables

**Figure 1 microorganisms-11-02039-f001:**
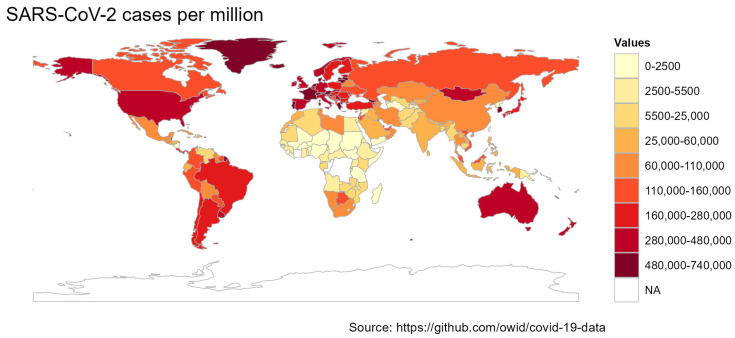
Frequency of confirmed COVID-19 cases per million across the world as of June 2023 [4].

**Figure 2 microorganisms-11-02039-f002:**
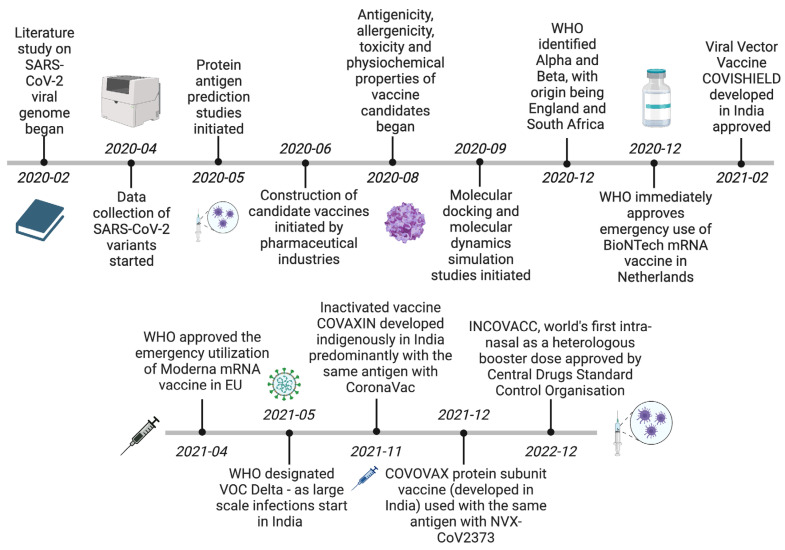
In silico vaccine development timeline [42,43].

**Figure 3 microorganisms-11-02039-f003:**
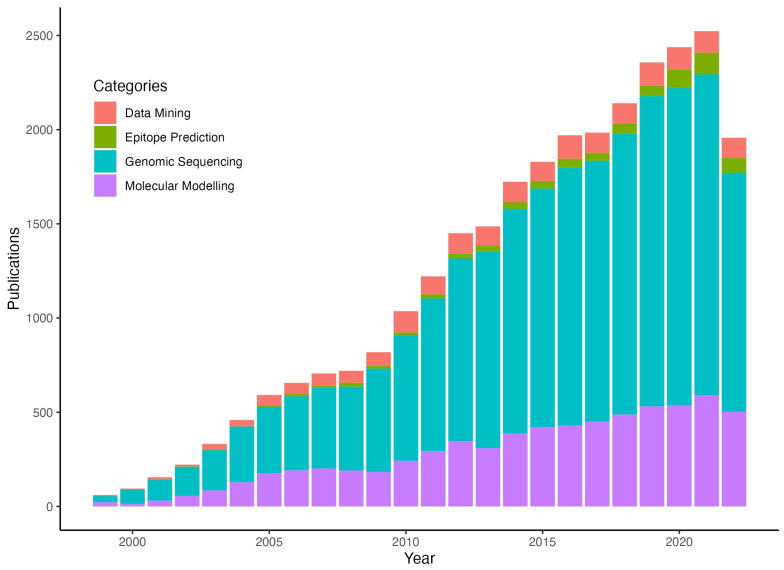
Stacked bar chart displaying the rising rate of published computational methods across various domains over the years.

**Table 1 microorganisms-11-02039-t001:** Categories with numerous bioinformatics tools have elucidated and assisted in the design, development, and deployment of potential vaccines in the fight against SARS-CoV-2 and its variants [27,39].

Areas Supporting Candidate Vaccine Development	Description	Examples
Genome Sequencing	Rapid processing and analysis of sequenced SARS-CoV-2 genomes aid in the rapid identification of mutations and thus aid in developing potential therapeutic targets and surveillance.	Trimmomatic, BWA, SAMTools, Seurat, GATK, SPAdes, Pangolin, IGV, Deseq2/EdgeR
Molecular Modeling	Various computational tools for molecular docking and molecular dynamics simulations facilitate the design and optimize therapeutic candidates.	PyMOL, Chimera, GROMACS, CHARMM, AMBER
Epitope Prediction	Numerous bioinformatics tools are being used to predict potential epitopes (antigenic determinants) that could be used to induce an immune response in the host and accelerate research/development.	NetMHC, IEDB, BepiPred, DiscoTope, Ellipro, ABCpred
Data Mining	Large-scale data mining and synthesizing of large-scale databases and information have speeded up the process of understanding virulence factors of new mutations, predicting their behavior, and creating appropriate drugs/vaccines to mitigate severe illness and curb spread, and inform public health policy.	Weka, Orange, KNIME, Cytoscape, TANGARA

## Data Availability

Please visit the following GitHub page https://github.com/ns012/dr-ns (accessed 2 July 2023) to find scripts utilized for construction of the plots as well as data used in the process.

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
