# Peer review of "COVID Variants, Villain and Victory: A Bioinformatics Perspective"

_microorganisms, 2023, doi:10.3390/microorganisms11082039_

Round 1

Reviewer 1 Report

Summary:

The authors evaluate the role of bioinformatics and next-generation sequencing techniques in refining diagnostic tools and expediting vaccine development for new SARS-CoV-2 variants. They highlight the crucial function of bioinformatics in battling COVID-19, enabling swift identification and understanding of new variants and facilitating rapid vaccine and therapeutic development. They underscore bioinformatics' significant contributions to critical biomedical fields. The core premise of this paper is sound, but some points still need to be addressed.

Major:

1.       A comprehensive review and revision of the section discussing the different variants of SARS-CoV2 could potentially uplift the overall quality of the manuscript. Specifically, the paragraph (line 52 - line 60) appears to need a rewrite to ensure the information is up-to-date. In addition, the inclusion of the references cited within this paragraph would be beneficial. Similarly, a re-evaluation and potential revision of the sentences (line 46 - line 49), which deal with the variants at the pandemic's peak, might be advantageous. Given that the manuscript was drafted a while back, the necessity for updates is understandable. Incorporating recent references into the manuscript, such as mentioning "as of June 2023", would bolster the article's relevance and precision.

2.       Figure 1 graphically represents the global frequency of COVID-19 cases per million. However, omitting a legend or key prevents the audience from understanding what the different colors signify.

3.       In Table 1, the authors have provided a list of bioinformatics tool categories, including genome sequencing, molecular modeling, epitope prediction, and data mining. It would be immensely beneficial to explore if any scaffolding tools or antibody omics tools could play a critical role in developing vaccines against SARS-CoV-2. Additional information about these tools could be incorporated into the paper. Moreover, I recommend enhancing Table 1 with an extra column that lists examples of the tools.

4.       Figure 2 presents the timeline of in-silico vaccine development. I highly recommend providing the references used to construct this figure.

5.       Figure 3 effectively displays the growing trend in the publication of computational methods. A consistent increase in the publication numbers for data mining, epitope prediction, and molecular modeling categories can be observed over the last 20 years (1999 – 2022). Conversely, there appears to be a notable reduction in genomic sequencing in 2022. It would be appreciated if you could elucidate the reasons behind this occurrence. Moreover, please provide more information on how to construct this figure.

Minor:

1.       Line 40: There appears to be an error in the manuscript. SIG uses four classifications, not the three mentioned by the authors.

2.       Line 52: It is advisable to use the abbreviation "VOC", as the term "variant of concern" has already been defined at line 42.

3.       Figure 2, in 2020-02, Literature "began"; in 2020-12, WHO "identifies" Alpha and Beta. Please revise these to the past tense "identified", since others are all past tense.

Moderate editing of English language required

Author Response

Summary:

The authors evaluate the role of bioinformatics and next-generation sequencing techniques in refining diagnostic tools and expediting vaccine development for new SARS-CoV-2 variants. They highlight the crucial function of bioinformatics in battling COVID-19, enabling swift identification and understanding of new variants and facilitating rapid vaccine and therapeutic development. They underscore bioinformatics' significant contributions to critical biomedical fields. The core premise of this paper is sound, but some points still need to be addressed.

Reviewer 1:

Major:

  1.       A comprehensive review and revision of the section discussing the different variants of SARS-CoV2 could potentially uplift the overall quality of the manuscript. Specifically, the paragraph (line 52 - line 60) appears to need a rewrite to ensure the information is up-to-date. In addition, the inclusion of the references cited within this paragraph would be beneficial. Similarly, a re-evaluation and potential revision of the sentences (line 46 - line 49), which deal with the variants at the pandemic's peak, might be advantageous. Given that the manuscript was drafted a while back, the necessity for updates is understandable. Incorporating recent references into the manuscript, such as mentioning "as of June 2023", would bolster the article's relevance and precision.

Response:

Thank you for bringing this matter to our attention. We genuinely appreciate your comment, and we fully agree with its validity. In response, we have diligently revised and updated the paragraph spanning from line 52 to line 60, incorporating the necessary changes. Moreover, we have included relevant references to further support the updated content. Additionally, we have revised the sentences from line 46 to line 49 and supplemented them with information regarding the dominant Variants of Interest (VOIs) currently circulating worldwide.

  1.       Figure 1 graphically represents the global frequency of COVID-19 cases per million. However, omitting a legend or key prevents the audience from understanding what the different colors signify.

Response:

Thank you for pointing this out. We agree with this comment, the legend has been included  in Figure 1. 

  1.       In Table 1, the authors have provided a list of bioinformatics tool categories, including genome sequencing, molecular modeling, epitope prediction, and data mining. It would be immensely beneficial to explore if any scaffolding tools or antibody omics tools could play a critical role in developing vaccines against SARS-CoV-2. Additional information about these tools could be incorporated into the paper. Moreover, I recommend enhancing Table 1 with an extra column that lists examples of the tools.

Response: 

Thank you for bringing this to our attention. We appreciate your comment, and we have taken it into consideration. We agree with your observation, and as a result, we have updated Table 1 to reflect the necessary changes.

Areas supporting candidate vaccine development:

Description

Examples

Genome Sequencing

Rapid processing and analysis of sequenced SARS-CoV-2 genomes aid in the rapid identification of mutations and thus aid in developing potential therapeutic targets and surveillance.

Trimmomatic, BWA, SAMTools, Seurat, GATK, SPAdes, Pangolin, IGV, Deseq2/EdgeR

Molecular Modeling

Various computational tools for molecular docking and molecular dynamics simulations facilitate the design and optimize therapeutic candidates.

PyMOL, Chimera, GROMACS, CHARMM, AMBER

Epitope Prediction

Numerous bioinformatics tools are being used to predict potential epitopes (antigenic determinants) that could be used to induce an immune response in the host and accelerate research/development

NetMHC, IEDB, BepiPred, DiscoTope, Ellipro, ABCpred

Data Mining

Large-scale data mining and synthesizing of large-scale databases and information have speeded up the process of understanding virulence factors of new mutations, predicting their behavior, and creating appropriate drugs/vaccines to mitigate severe illness and curb spread, and inform public health policy.

Weka, Orange, KNIME, Cytoscape, TANGARA

  1.       Figure 2 presents the timeline of in-silico vaccine development. I highly recommend providing the references used to construct this figure.

Response: Thank you for pointing this out, we have added the key references for creating Figure 2 which included insight from the following paper(s): 

  1. Sumon, Tofael Ahmed, et al. "A revisit to the research updates of drugs, vaccines, and bioinformatics approaches in combating COVID-19 pandemic." Frontiers in Molecular Biosciences 7 (2021): 585899.
  2. Kumar, V.M., Pandi-Perumal, S.R., Trakht, I. et al. Strategy for COVID-19 vaccination in India: the country with the second highest population and number of cases. npj Vaccines 6, 60 (2021). https://doi.org/10.1038/s41541-021-00327-2
  1.       Figure 3 effectively displays the growing trend in the publication of computational methods. A consistent increase in the publication numbers for data mining, epitope prediction, and molecular modeling categories can be observed over the last 20 years (1999 – 2022). Conversely, there appears to be a notable reduction in genomic sequencing in 2022. It would be appreciated if you could elucidate the reasons behind this occurrence. Moreover, please provide more information on how to construct this figure.

Response: Thank you for bringing this to our attention. We greatly appreciate your comment, and we agree with your observation. As a result, we have made the necessary adjustment by including an explanation for the observed trend within the caption of Figure 3.

This figure was constructed in R with the following steps : 

  1. Data was downloaded from pubmed against the query terms : “genomic sequencing”, “molecular modeling”, “epitope prediction”, and “data mining”. The data was exported in CSV form and aggregated in wide form with a year column, followed by the relevant count of the research papers for each year mentioned against the queried term. 
  2. Data was loaded into Rstudio (R version 4.1.1), preprocessed and structured. 
  3. Using ggplot2 library - a stacked barplot was made. Appropriate customizations were done for generating the plot. 

References for the trend have been also included: 

  1. Harper, L., et al. "The impact of COVID-19 on research." Journal of pediatric urology 16.5 (2020): 715.
  2. Venkatesh, Viswanath. "Impacts of COVID-19: A research agenda to support people in their fight." International journal of information management 55 (2020): 102197.

Minor:

  1.       Line 40: There appears to be an error in the manuscript. SIG uses four classifications, not the three mentioned by the authors.

Response:We sincerely appreciate your feedback, and we have promptly addressed the mistake and made the necessary correction.

  1.       Line 52: It is advisable to use the abbreviation "VOC", as the term "variant of concern" has already been defined at line 42.

Response: We have corrected this mistake.

  1.       Figure 2, in 2020-02, Literature "began"; in 2020-12, WHO "identifies" Alpha and Beta. Please revise these to the past tense "identified", since others are all past tense.

Response: Thank you for pointing this out. We have corrected this mistake.

Reviewer 2 Report

Authors present an interesting overview of bioinformatics tools. Paper is well written and conclusion are sound. I have some suggestion for the authors.

1)  The variants under this group includes Apha (B.1.1.7) from United 54 Kingdom with ~50% infection rate, Beta (B.1.351) from South Africa with ~50% infection 55 rate, Delta (B.1.617.2)

Why authors do not include also Omicron?

2) Bioinformatics has created a milestone and has played an essential role in COVID-19 81 research since the emergence of pandemic [8], especially in detecting and tracking new 82 variants as well as allowing for open data sharing across the world. Wide varieties of bi- 83 oinformatics tools and techniques have been used successfully in interpretation of the 84 genomic architecture of SARS CoV-2 and its variants.

2) Bioinformatics tools have also been used to model spread and to design containment measures and in general to support clinical researcu. Please add some references (not limited to) as Hiram Guzzi, Pietro, Francesco Petrizzelli, and Tommaso Mazza. "Disease spreading modeling and analysis: A survey." Briefings in Bioinformatics 23.4 (2022): bbac230.Kumar Das, Jayanta, et al. "Data science in unveiling COVID-19 pathogenesis and diagnosis: evolutionary origin to drug repurposing." Briefings in Bioinformatics 22.2 (2021): 855-872.

3) Discussion and Conclusion: please discuss the evolution considering the declaration of the end of the pandemic.

English is fine

Author Response

Authors present an interesting overview of bioinformatics tools. Paper is well written and the conclusions are sound. I have some suggestions for the authors.

1)  The variants under this group includes Apha (B.1.1.7) from United 54 Kingdom with ~50% infection rate, Beta (B.1.351) from South Africa with ~50% infection 55 rate, Delta (B.1.617.2)

Why do authors not include Omicron?

Response: Thank you for pointing this out. We agree with this comment, thus have discussed Omicron and its diversity, as well as WHO changing their classification system for Omicron.

2) Bioinformatics tools have also been used to model spread and to design containment measures and in general to support clinical researcu. Please add some references (not limited to) as Hiram Guzzi, Pietro, Francesco Petrizzelli, and Tommaso Mazza. "Disease spreading modeling and analysis: A survey." Briefings in Bioinformatics 23.4 (2022): bbac230.Kumar Das, Jayanta, et al. "Data science in unveiling COVID-19 pathogenesis and diagnosis: evolutionary origin to drug repurposing." Briefings in Bioinformatics 22.2 (2021): 855-872.

Response:Thank you for bringing this to our attention. We truly appreciate your input. In response, we have taken immediate action and added relevant references that specifically discuss the crucial role of bioinformatics and data science in supporting clinical research and informing public health policy decisions.

3) Discussion and Conclusion: please discuss the evolution considering the declaration of the end of the pandemic.

Response: Thank you for pointing this out. We agree with this comment and have added a section discussing the potential evolution path of COVID-19 and the end of the pandemic.

Round 2

Reviewer 1 Report

I noticed a faint line beneath Figure 3 (line 285), which seems to be present after exporting from your software. Could you please ensure that the line is cropped before publishing?

N/A

Author Response

Thank you for your valuable feedback. We have taken your comments into consideration and made the necessary improvements. The suggested image has been regenerated and meticulously cropped to meet your requirements. We appreciate your input and hope that the revised image meets your expectations.